# Spatial attention enhances network, cellular and subthreshold responses in mouse visual cortex

Anderson Speed [1], Joseph Del Rosario[1], Navid Mikail[1] & Bilal Haider[1]*

Internal brain states strongly modulate sensory processing during behaviour. Studies of visual processing in primates show that attention to space selectively improves behavioural and neural responses to stimuli at the attended locations. Here we develop a visual spatial task for mice that elicits behavioural improvements consistent with the effects of spatial attention, and simultaneously measure network, cellular, and subthreshold activity in primary visual cortex. During trial-by-trial behavioural improvements, local field potential (LFP) responses to stimuli detected inside the receptive field (RF) strengthen. Moreover, detection inside the RF selectively enhances excitatory and inhibitory neuron responses to task-irrelevant stimuli and suppresses noise correlations and low frequency LFP fluctuations. Whole-cell patch-clamp recordings reveal that detection inside the RF increases synaptic activity that depolarizes membrane potential responses at the behaviorally relevant location. Our study establishes that mice display fundamental signatures of visual spatial attention spanning behavioral, network, cellular, and synaptic levels, providing new insight into rapid cognitive enhancement of sensory signals in visual cortex.

[1] Georgia Institute of Technology & Emory University, Atlanta, GA, USA. *email: bilal.haider@bme.gatech.edu

Mice have become an important tool for understanding behavioral modulation of sensory processing in mammalian cortical circuits. Studies in awake mice have revealed that locomotion[1–3], shifts of arousal[4–6], motor activity[7], and behavioral tasks[8–12] modulate sensory responses. These and other studies in transgenic mice have further uncovered the role of cortical neuronal subtypes[13,14] during sensory response modulation. However, it remains unknown whether cortical circuits in mice selectively enhance sensory responses to stimuli in discrete regions of space, as classically observed in primates during selective attention tasks.

Fundamental studies in primates outline several ways spatial attention modulates behavior and visual processing. These include selective increases in spiking to stimuli at a behaviorally relevant location[15–19], reduction of correlated neural variability[20,21], selective modulation of low- and high-frequency synchronization[22–24], and activity modulation in specific cortical layers[25]. These observations have inspired theories of biophysical mechanisms underlying attentional modulation[26–28], but these have not yet been tested across behavioral, network, cellular, and synaptic levels, largely owing to the challenge of performing all of these combined measurements in primates.

Establishing selective attention tasks in mice provides opportunity for detailed study of neural circuits underlying cognitive modulation of sensory processing. Mice can indeed use cues to selectively attend to one sensory modality versus another[29]. Moreover, a recent study showed that visual spatial cueing improves ensuing detectability of stimuli at the cued location[30], consistent with effects of spatial orienting and attention[18,31,32]; crucially, this study did not examine the cortical basis for this behavior, did not record neural activity in visual cortical areas, or directly assess if cortical dynamics in mice resemble those in the primate visual system during spatial attention tasks.

Here, we show that a visual detection task with spatial context elicits behavioral improvements in mice consistent with the effects of spatial attention. We simultaneously performed high-density silicon probe recordings in the primary visual cortex (V1), and found that behavioral improvements were accompanied by stronger neural responses across network and cellular levels, particularly in layer 4 (L4). These observations were not explainable by several non-sensory or nonspatial factors. Larger neural responses were accompanied by reductions in both pairwise noise correlations and low-frequency network activity. Moreover, whole-cell patch–clamp recordings revealed enhanced subthreshold responsiveness at the behaviorally relevant spatial location. This work establishes that fundamental neural correlates of visual spatial attention observed in primates also operate in mice during selective improvements of visual spatial behavior. Our findings provide new insight into the cellular, network, and synaptic basis for rapid modulation of early sensory processing during spatial behaviors.

## Results

### Spatial context improves trial-by-trial detection performance.
To investigate the cellular and network mechanisms underlying spatial contextual modulation, we developed a visual task where mice reported detection of spatially localized stimuli (Gabor gratings, σ = 10°) by licking for water reward. Stimuli appeared only after a mandatory period of no licking (0.5–6 s, randomized per trial), and rewards were only available upon first lick during the stimulus (1–1.5 s duration). Stimuli appeared with unpredictable timing, but always at one of two discrete locations of visual space (either in the binocular or monocular visual fields; Fig. 1a). Stimuli appeared at a range of contrasts (Fig. 1b; Supplementary Fig. 1) at one of these spatial locations for the

duration of a block of ~25 repeated trials; then, without warning, stimuli switched to the other spatial location for a new block of trials (Fig. 1d). Thus, stimulus location remained spatially fixed within a block of trials, yet alternated locations across blocks of trials. This spatial context allowed us to investigate if sequential trials at a fixed location elicited behavioral effects consistent with spatial attention.

Indeed, detection was slowest and least accurate on the first trial within the block, when stimulus location was not predictable from the previous trial; thereafter, detection speed and accuracy significantly improved on consecutive trials at the same location (Fig.1e, f; time constant of improvement ~5 trials). This increase in performance across trials was associated with a progressive increase in hit rates, and an overall decrease in false alarm rates. Consequently, detection sensitivity (d') improved significantly within five trials (Fig. 1f; from 1.1 ± 0.1 to 1.6 ± 0.1; mean ± s.e.m.; $p < 0.001$, Wilcoxon rank-sum test). Criterion also decreased significantly from first (0.37 ± 0.04) to fifth trials (0.22 ± 0.03; $p < 0.001$, rank-sum test), reflecting that mice tended to respond more frequently with hits on trials later in the block as detection sensitivity improved. Critically, behavioral performance required spatially localized activity in V1: optogenetically inactivating monocular primary visual cortex (V1) selectively abolished monocular detection (Fig. 1c; Supplementary Fig. 1).

### Neural response amplitude increases with improved detection.
During these behavioral improvements, neural responses in V1 strengthened. We recorded laminar local field potentials (LFP; Supplementary Fig. 2) in monocular V1 (Supplementary Fig. 4), and observed a significant amplification of Layer 4 (L4) LFP responses to monocular stimuli, with the same time constant as the simultaneous behavioral improvements (Fig.1 g–j; 72% increase in LFP response from Trial 1 to 5 for correct detection trials). This occurred on the earliest portion of the sensory response across all cortical layers, but most prominently in L4 (Supplementary Fig. 2). These V1 responses strengthened in the exact same mice and on the exact same trials in which behavior improved (Fig. 1e, f).

### Behavioral improvement is not due to arousal or motivation.
Changes in arousal, eye position, or motivation did not explain behavioral improvements. We measured pupil area as a proxy for global changes of arousal. Pupil area prior to stimulus onset remained remarkably constant from first to fifth correct detection trials (Fig. 2a, <0.1% change from mean). These changes were tenfold smaller than those preceding correct versus incorrect trials (4 ± 13%, see Speed et al.[12]). Pupil position also remained stable, deviating <2° from first to fifth trials within the block (Fig. 2c); these deviations were tenfold smaller than the extent of the visual stimuli (Fig. 2d). Next, we reasoned that changes in motivation across trials would manifest as measureable changes in licking. If motivation for reward increased from first to fifth trials, this should drive impulsivity[33] and increase premature stray licks preceding stimulus onset; likewise, consummatory licks should increase across sequential rewards, if motivation grew. However, neither of these scenarios were evident: there was no increase in stray "anticipatory" licking preceding stimulus onset, and no increase in consummatory lick vigor on rewards from first to fifth trials (Supplementary Fig. 3). Taken together, these results suggest that increased arousal or motivation were not apparent from simultaneous measures of pupil and licking, and neither co-varied with the time course of behavioral improvements. Instead, spatial context progressively improved stimulus detection speed and accuracy, while the simultaneously measured neural responses strengthened.

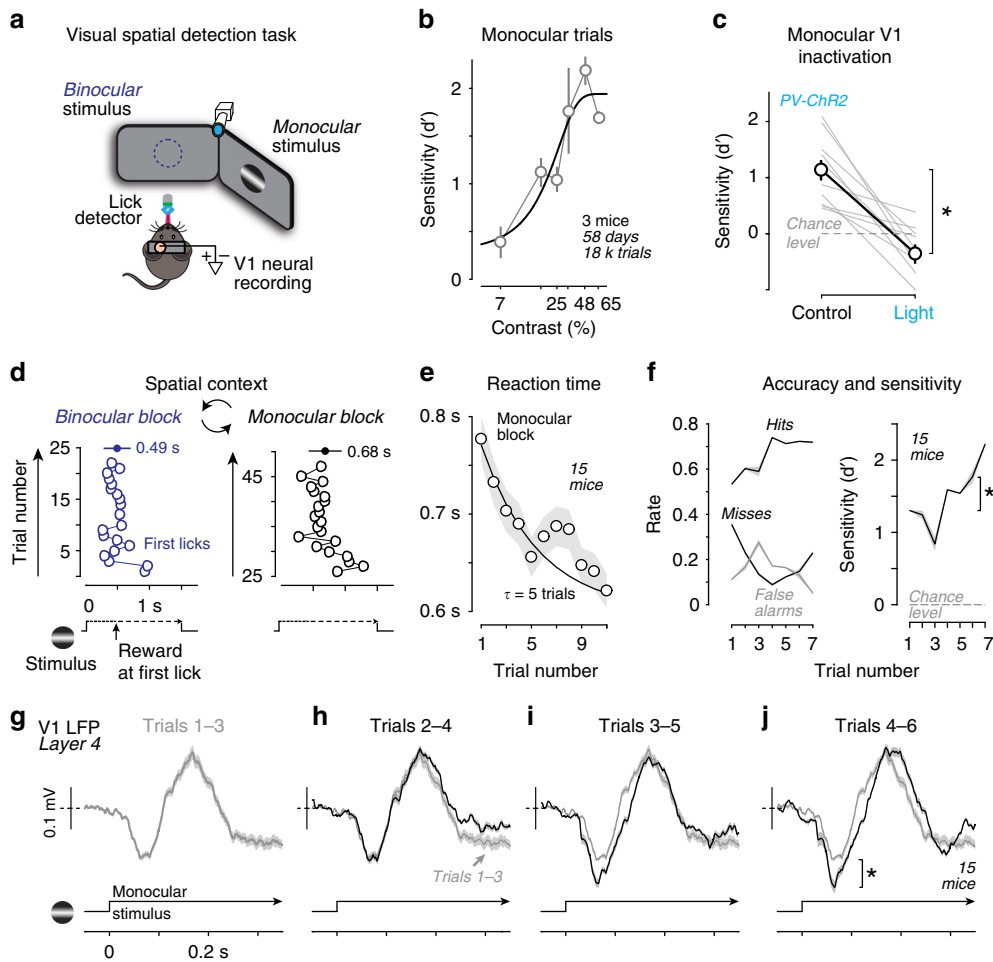

**Fig. 1 Spatial context improves behavior and strengthens LFP responses. a** Visual spatial detection task. Mice were head-fixed and stationary in a semi-enclosed plastic tube. Mice faced two computer monitors and performed the task during simultaneous measurements of licking, pupil, and neural activity. Stimuli appeared in either binocular or monocular visual space. **b** Monocular detection sensitivity (d') increases as a function of contrast ($n = 3$ mice, 58 behavioral sessions, >18,000 trials). These mice were trained extensively at the exact same contrast ranges, and are part of the population analyzed throughout the study. Dark curve shows psychometric fit. **c** Optogenetic inhibition of monocular V1 (interleaved on 25% of trials) significantly impaired monocular detection sensitivity (d', 1.13 ± 0.19 to −0.36 ± 0.18; mean ± SD; $p < 0.01$; Wilcoxon signed-rank test; $n = 10$ days and 1575 trials). During inactivation, d' was not significantly different from chance level ($p < 0.01$, sign test). Data from two PV-cre x A1-32-ChR2 mice. See also see Supplementary Fig. 1. **d** Example session of visual spatial detection during neural recordings. Stimuli (gratings) appear in binocular (blue) or monocular (black) visual fields, in blocks of consecutive trials at each location. Blocks alternate repeatedly within session (after 25 trials in this example). Rewards delivered upon first lick (circles, correct detection) during stimulus window. Reaction times: binocular, 0.49 ± 0.08 s; monocular, 0.68 ± 0.1 s. Mean ± s.e.m. **e** Reaction time improves significantly across consecutive trials within block (15 mice; 143 blocks; $p < 0.05$, Wilcoxon rank-sum test, throughout figure). Mean ± s.e.m., monocular trials. Time constant (τ) of improvement = 4.5 trials (exponential fit). **f** Detection accuracy and sensitivity improve significantly across same trials as in **e**. Left, hit rate increases 27% ($p < 0.001$, rank-sum test), miss rate decreases 50% ($p < 0.001$), false alarm rate decreases 16% ($p = 0.4$) from Trials$_{(1-3)}$ to Trials$_{(4-6)}$. Right, psychometric sensitivity (d') increases significantly across same trials (Trial$_{(1-3)}$:1.1 ± 0.1; Trials$_{(4-6)}$: 1.6 ± 0.1; mean ± s.e.m.; $p < 0.001$, rank-sum test). **g**–**j** Local field potential (LFP) responses in Layer 4 of primary visual cortex (V1) during same behavioral sessions as **e** and **f**. LFP response increases significantly from trials 1 to 3 (**g**, gray) to trials 4 to 6 (**j**, black; 72% increase; [Trials$_{(4-6)}$ Trial$_{(1-3)}$]/Trial$_{(1-3)}$; $p < 0.01$, sign test). Mean ± s.e.m. across correct trials. Dashed line at 0 mV. Fifteen mice, 21 recording sessions in monocular V1 (see Supplementary Fig. 4).

These observations in mice are consistent with well-established effects of selective visual spatial attention[18,30,31].

**Neural response enhancements are spatially specific**. We hypothesized that if response enhancement was due to spatial context, then it should also selectively enhance responses to task-irrelevant stimuli at the same behaviorally relevant locations[15,16]. We therefore presented task-irrelevant probe stimuli (vertical bars 10° wide, ±5% contrast from gray screen) during the intertrial intervals separating successive presentations of the grating stimuli detected for rewards (Fig. 3a). Brief probes (0.1 s duration, 0.3 s

interval) appeared one at a time in a randomly selected location anywhere in the binocular or monocular visual fields (see the Methods section). Since the receptive fields (RF) of our recordings were in monocular V1 (Supplementary Fig. 4), stimulus detection occurred inside the RF on monocular blocks and, conversely, outside the RF during blocks at the binocular location. This design allowed us to compare neural responses to the very same task-irrelevant probe stimuli appearing inside the RF in two different spatial contexts in interleaved alternating blocks (Fig. 3a, left versus right). Since probe stimuli were faint, brief, and task-irre-levant, they did not elicit any behavioral response: following probe onset inside the RF there was no increase in licking, and stimulus

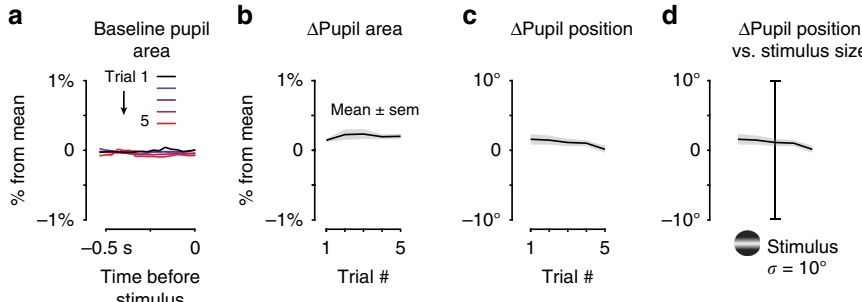

**Fig. 2 Pupil dynamics do not explain behavioral improvements. a**, **b** Average pupil area before stimulus onset does not change from first to fifth trial (0.03 ± 0.17% versus 0.04 ± 0.07%, p = 0.10). In total, 426 correct monocular detection blocks in 22 sessions. **c** In the same trials, pupil position before stimulus onset changes <2° (Trial 1, 1.6 ± 3.6° versus 0.14 ± 2.4°, relative to mean position across entire session; *p = 0.02*, Wilcoxon signed-rank test). **d** Changes of pupil position (<2°) are an order of magnitude smaller than spatial extent of detected stimuli (grating, 2σ = 20°).

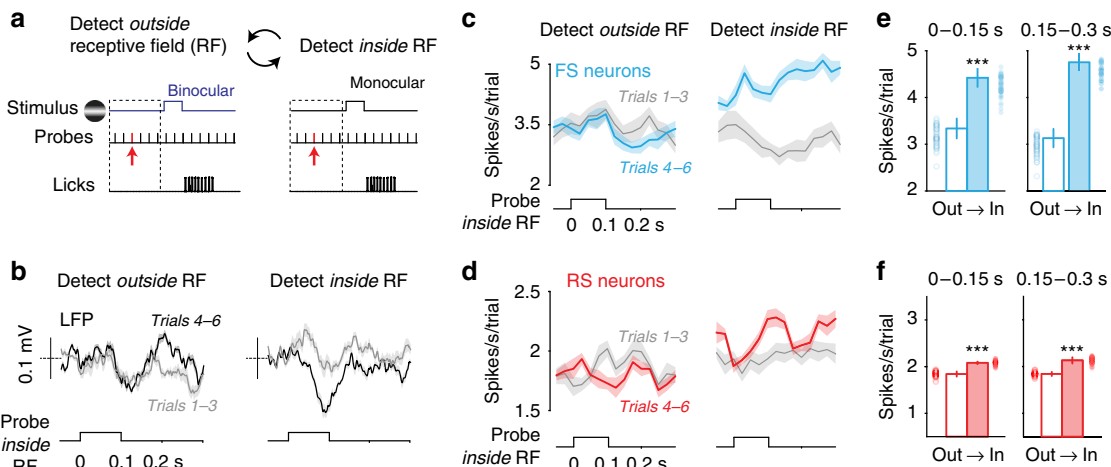

**Fig. 3 Spatially selective enhancement of LFP, excitatory and inhibitory spiking. a** Task-irrelevant probe stimuli (flashed bars, 0.1 s duration) appear randomly across the entire visual field during spatial detection trials. In monocular V1, binocular detection trials elicit attention outside the receptive field (RF), while monocular detection trials elicit attention inside the RF. Responses to identical probes (red arrow) computed during the interval preceding correct detection, and compared across stimulus blocks. **b** Left, during detect outside RF trials, LFP responses to probes inside the RF are not different on Trials(1–3) (gray) versus Trials(4–6) (black). Right, during detect inside RF trials, LFP responses to same probes are significantly enhanced across consecutive trials (79 ± 38%; p < 0.05, Wilcoxon rank-sum test, throughout figure). Mean ± s.e.m. (15 mice, 21 recording sessions from monocular V1). **c** Same sessions as **b**, for spikes from fast spiking (FS) putative inhibitory neurons (n = 67, all layers). Detection inside the RF increased FS probe responses by 1.1 ± 0.4 spikes per s per trial (p < 0.001, rank-sum test). Mean ± s.e.m. **d** Same as **c**, for spikes from regular spiking (RS) putative excitatory neurons (n = 225, all layers). Detection inside the RF increased RS probe responses by 0.2 ± 0.1 spikes per s per trial (p < 0.001, rank-sum test). Mean ± s.e.m. **e** FS neurons show enhanced probe responses for both early (<0.15 s; out: 3.3 ± 0.2 vs in: 4.4 ± 0.2 spikes per second, p < 0.001 rank-sum test) and late sensory activity (0.15–0.3 s; out: 3.1 ± 0.2 vs in: 4.8 ± 0.2 spikes per second, p < 0.001, rank-sum test). Late activity significantly greater than early on detect inside RF trials (p < 0.05, rank-sum test). **f** Same as **e**, for RS neurons early (out: 1.8 ± 0.06 vs in: 2.08 ± 0.05 spikes per second, p < 0.001, rank-sum test), late (out: 1.8 ± 0.04 vs in: 2.13 ± 0.07 spikes per second, p < 0.001, rank-sum test). Late activity significantly greater than early on detect inside RF trials (p < 0.05, rank-sum test). ***p < 0.001.

detection accuracy was unaffected (Supplementary Fig. 5). Probe stimuli inside the RF were thus free from confounds of motor planning or reward association[34–36], enabling isolated measurement of the impact of spatial context on sensory processing inside the RF.

Detection trials inside the RF significantly and selectively enhanced LFP responses to probe stimuli in L4 (Fig. 3b); responses to these same probes did not change across trials during detection outside of the RF. Enhancement of the LFP response developed gradually over the course of each block, and with the same time constant as behavioral and neural response enhancement to detected stimuli (cf. Fig. 1). Probe responses tended to be strongest at sites where neuronal RFs were well-aligned with the detected stimulus location (85% greater modulation when RF location was

<10° away from center of detected grating stimulus, versus >10° away, p = 0.07, rank-sum test).

**Enhanced neural responses to probes are not due to arousal.** Arousal did not explain enhanced neural responses inside the RF (Fig. 4). We again used pupil area to segregate trials by arousal level, and found that during detection inside the RF, probes evoked similar amplitude LFP responses in low or high arousal trials; in both cases, LFP responses sorted by arousal level were much smaller than the enhanced LFP responses on the fifth trial of detection inside the RF. This spatially selective enhancement of responses to task-irrelevant stimuli mirrors effects of spatial attention in the primate visual system[16,37].

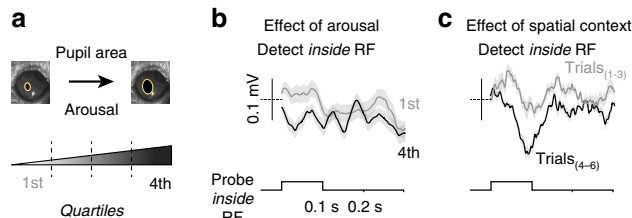

**Fig. 4 Effect of arousal is not comparable to that of spatial context. a** Pre-stimulus pupil area preceding each trial of correct detection split into quartiles. Larger pupil area indicates greater arousal. **b** LFP responses to probes presented inside the receptive field (RF), on detect inside RF trials. Gray indicates population LFP responses on trials in 1st quartile of pupil area (low arousal), black indicates responses on trials in 4th quartile of pupil area (high arousal). Mean ± s.e.m. **c** LFP probe responses during same trials as in **b**, separated by Trials(1–3) (gray) versus Trials(4–6) (black) in block (replotted from Fig. 3b).

**Enhancement of excitatory and inhibitory neuron responses.**
Detection inside the RF increased excitatory and inhibitory neuron spiking to probes. We classified neurons across layers as either regular spiking (RS) putative excitatory neurons or fast spiking (FS) putative parvalbumin (PV) inhibitory neurons[1,12,38] (Supplementary Fig. 6). Detection inside the RF significantly enhanced RS and FS spiking responses to probes inside the RF from the first to fifth trials (Fig. 3c, d); again, there was no change in responses to these same probes across trials with detection outside the RF. FS neurons showed greater enhancement (32 ± 0.01%) than RS neurons (13 ± 0.0004%). Like the LFP, sensory response enhancements were prominent at short latency (<0.15 s) and slightly but significantly larger at longer latencies (0.15–0.3 s), perhaps owing to delayed and prolonged responses to low stimulus contrast[39], or to spatially specific but sustained increases in neural excitability resulting from improved behavioral performance across trials. By comparing this spatial contextual modulation of probe responses to contrast response functions (Supplementary Fig. 9), we found that RS and FS response enhancement was more consistent with contrast gain[40] than response gain[41] (Supplementary Fig. 10).

**Stimulus expectation alone does not explain RS or FS spiking.**
Increased spiking during detection inside the RF was not due to stimulus expectation. We used zero contrast trials (blanks) to identify if non-sensory factors related to stimulus expectation increased firing rates. Blank trials were randomly interleaved throughout the behavioral sessions (20% of total trials), and occurred with the same intertrial interval statistics as the grating stimuli detected for rewards (see the Methods section). This allowed us to isolate firing rates both preceding and immediately following the "expected" appearance of a visual stimulus across trials, but in the absence of any sensory-evoked activity. Indeed, there was no increase in RS or FS firing rates preceding or following the onset of a blank stimulus on first or fifth trials within the block (Supplementary Fig 7). We further separated blank trials into those where mice licked (false alarms), and compared with trials with no licks (correct rejects). Again, there was no evidence for increased firing rates in FS or RS cells on blank trials with ensuing licks, as could be expected with motor readiness for reward. Similarly, we found no evidence for firing rate increases on blank trials sorted for detection inside versus outside the RF (Supplementary Fig. 8, FS detect outside RF: 4.32 ± 1.2; detect inside RF: 2.6 ± 0.7 spikes per second, $p = 0.6$; RS detect outside RF: 1.9 ± 0.2; detect inside RF: 1.9 ± 0.2 spikes per second, $p = 0.7$, signed-rank test). These results argue against the possibility that non-sensory stimulus expectation increased firing rates, and instead provide support for the specific interaction between spatial context and the enhancement of sensory-driven activity inside the RF.

**Spatial context reduces noise correlations inside the RF.**
Detection inside the RF selectively reduced pairwise noise correlations during probe responses[20] (detect outside RF: 0.021 ± 0.001; detect inside RF: 0.017 ± 0.001; $n = 1500$ pairs, aggregated across cell types and layers; $p < 0.01$, rank-sum test). Although detection inside the RF decreased noise correlations between pairs of RS neurons or pairs of FS neurons, this was most evident in pairwise interactions between FS and RS neurons[21] (Fig. 5). This aspect of spatial contextual modulation in mouse V1 is broadly consistent with primate studies of selective visual attention[42,43].

**Spatial context reduces low-frequency activity inside the RF.** At these same recording sites, detection inside the RF reduced low-frequency LFP power across trials (Fig. 6). We calculated LFP power spectral density during the detected stimulus on single trials, and found that LFP power in low-frequency bands (<10 Hz) significantly decreased from first to fifth trials of detection inside the RF, in both L4 (Fig. 6; modulation index (MI): −0.07 ± 0.06, $p < 0.05$, Wilcoxon signed-rank test), and also in L5/6 (MI = −0.12 ± 0.05, $p < 0.05$, signed-rank test), with no significant modulation in L2/3 (MI = −0.09 ± 0.09, $p = 0.70$, signed-rank test). We also examined stimulus-driven modulation of gamma frequencies, but this was not significantly different across successive trials of detection inside the RF. This was not because the stimulus did not evoke high-frequency activity: LFP gamma power was significantly larger for stimuli appearing inside versus outside the RF (24.6 ± 4.0 dB inside RF vs 24.0 ± 5.1 dB outside RF, $p < 0.05$, signed-rank test).

**Spatial context elicits membrane potential depolarization.**
Since LFP in awake mouse V1 directly reflects subthreshold excitatory and inhibitory synaptic activity[44], we hypothesized that detection inside the RF operates directly on subthreshold membrane potential ($V_m$). Based on our previous simultaneous recordings of LFP and $V_m$ in awake V1[44], we predicted directly from the LFP responses (Supplementary Fig. 11) that detection inside the RF should depolarize membrane potential responses to probes by ~1 mV (Fig. 7a).

We then directly measured subthreshold $V_m$ modulation during shifts of spatial context. We performed whole-cell patch–clamp recordings[45,46] during task performance (Fig. 7b), and analyzed $V_m$ responses to probes appearing inside the RF (Supplementary Fig. 11), while spatial context alternated outside or inside the RF across blocks. Strikingly, detection inside the RF evoked depolarization with two components: a tonic depolarization of 4.9 ± 0.4 mV, and a transient probe-evoked depolarization of 2.2 ± 0.3 mV (Fig. 7c; $p < 0.001$ for both, signed-rank test). Consistent with depolarization resulting from increased synaptic activity, spontaneous membrane potential fluctuations ($V_m$ SD) slightly but significantly increased during blocks of detection inside versus outside the RF (Fig. 7d; 2.5 ± 0.1 vs 2.1 ± 0.1 mV; $p = 0.01$, signed-rank test). Based upon the measured power law relationship between $V_m$ and spike rate in excitatory neurons, 1–2 mV of synaptic depolarization produced small but measurable increases in spike rate (Supplementary Fig. 11), as we recorded in RS neurons (Fig. 7d). These data reveal that spatial context selectively and directly increases subthreshold synaptic activity and membrane depolarization in V1 during behavioral improvements, and in parallel with local increases in excitation and inhibition. These measurements in mice provide verification of long-theorized mechanisms[18,26,47,27] of response modulation proposed to underlie attentional enhancements.

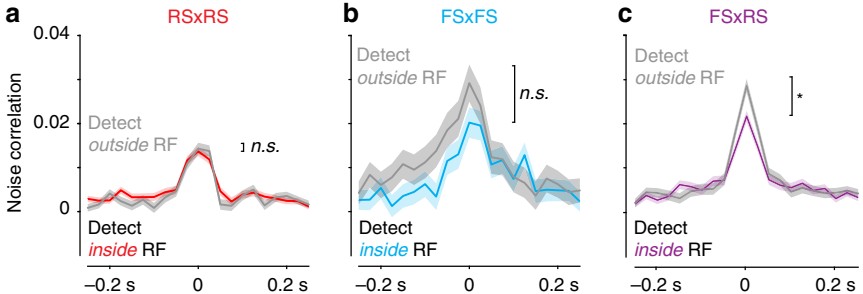

**Fig. 5 Spatial and cell-type-specific reduction in noise correlations. a** Noise correlations between pairs of regular spiking (RS) neurons during probe stimulus presentation ($n = 797$ pairs). Correlations for detect inside RF trials (red) versus detect outside RF (gray). Mean ± s.e.m. No significant difference in peak noise correlations (detect outside RF: 0.0143 ± 0.001; detect inside RF: 0.0137 ± 0.001; $p = 0.98$, Wilcoxon rank-sum test throughout figure). **b** As in **a**, between pairs of fast spiking (FS) neurons during stimulus presentation ($n = 95$ pairs). No significant difference in peak noise correlations (detect outside RF: 0.029 ± 0.004; detect inside RF: 0.020 ± 0.003; $p = 0.23$, rank-sum test). **c** As in **a**, between pairs of regular spiking (RS) and FS neurons during stimulus presentation ($n = 608$ pairs). Significant reduction in peak noise correlations during trials of detection inside RF (detect outside RF: 0.029 ± 0.002; detect inside RF: 0.022 ± 0.001; $p < 0.001$, rank-sum test).

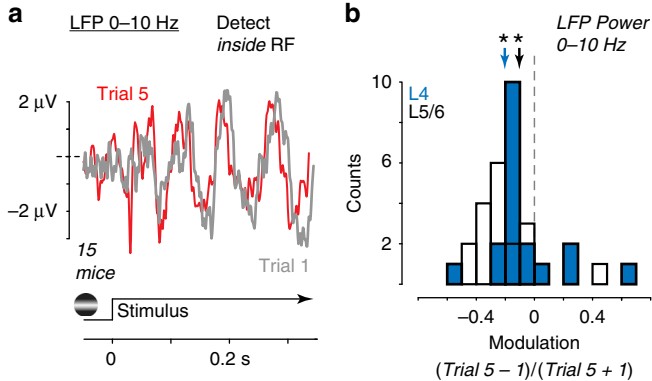

**Fig. 6 Reduced 0–10 Hz LFP power on trials with behavioral improvements. a** Average stimulus-evoked local field potential (LFP) responses filtered from 0 to 10 Hz, during same behavioral and recording sessions as Fig. 1 g–j. Trial 1 (gray) vs trial 5 (red), 15 mice, 21 recording sessions. **b** L4 and L5/6 exhibited significant suppression of 0–10 Hz LFP power across successive detection trials inside the RF (L4: MI = −0.07 ± 0.06, $p < 0.05$, L5/6: MI = −0.12 ± 0.05, $p < 0.05$, signed-rank test).

Taken together, we found that spatial context enhanced behavior and sensory signals across network, cellular, and subthreshold synaptic levels in mouse V1 (Fig. 8). In contrast to recordings in primate visual cortex[48], contextual modulation in mouse V1 was larger in LFP than in RS or FS firing, even though all were recorded from the same sites during the same behavioral trials (Fig. 3e–j; 79% for LFP, 32% for FS neurons, 13% for RS neurons; $p < 0.05$; nonparametric ANOVA). Consistent with this large LFP modulation, subthreshold $V_m$ was enhanced to an even greater extent, with transient probe-evoked depolarization increasing by 108% during detection trials inside the RF.

## Discussion

Our study reveals that mice display several fundamental signatures of visual spatial attention, as classically observed in primates. These signatures span behavioral, network, cellular, and synaptic levels, and together provide new insight into local mechanisms underlying rapid cognitive modulation of sensory signals in V1. Establishing spatial attentional effects across behavioral and neural levels represents a key step forward for studies of mouse vision and cognition. Our findings complement and extend others[30] in countering the long-standing view that mice are poorly suited for

study of neural substrates of selective visual attention, due to lack of high-resolution foveal vision, absence of visually guided saccades, lack of orientation columns, and limited cognitive capacity that may preclude training in attention-demanding tasks[49]. Our study establishes a paradigm wherein mice selectively improve behavior using spatial context, and this simultaneously enhances sensory responsiveness specifically at the behaviorally relevant spatial location. The consistency of our framework with prior studies in primates aids and anchors interpretability of our findings, discussed in turn below.

Our study used a visual detection task with a block design, where the detected stimulus itself provided spatial context across successive trials. Behavioral improvements were gradual, with a time constant of five trials. These improvements in reaction time and accuracy are consistent with a recent study in mice that explicitly conveyed spatial salience with a Posner cue[30]. One advantage of our design is the lack of a cueing stimulus inside the RF immediately preceding the detected stimulus; multiple stimuli drive complex neural responses[50] that pose greater challenges for interpretation as compared with our single-stimulus design. A second advantageous feature of our design is that mice were not running, minimizing complicated interactions between visual responses, locomotion, and arousal. A third advantage of our task is that the interleaved trials of monocular detection are more difficult than binocular detection[12], perhaps accentuating the magnitude and visibility of behavioral and neural effects of spatial context[51]. Our results bear some similarities with neural correlates of visual selection in mice[52]. Direct comparison of our task with other head-fixed visual spatial tasks for mice will be an important avenue for investigation.

At the network level, detection inside the RF amplified LFP responses across all layers of V1, but particularly in L4. Spatial attention also modulates LFP response amplitude in V4 of primates, but to a lesser degree than observed here, and with suppression and enhancement of early and late LFP phases[48]. In our study, we also observed a reduction of low-frequency LFP power at the same sites showing LFP amplitude increases, in both L4 and L5/6. Several primate studies have shown reduction of low-frequency fluctuations with spatial attention, both in the LFP[23,53,54] and also with single and multiunit spike coherence with LFP[21,24,25]. Reductions in low-frequency activity are particularly beneficial for stimulus encoding[55], and causally related to attentional improvements[56]. An unexpected difference between primate studies and our findings was lack of modulation of gamma oscillations during subsequent trials of detection inside the

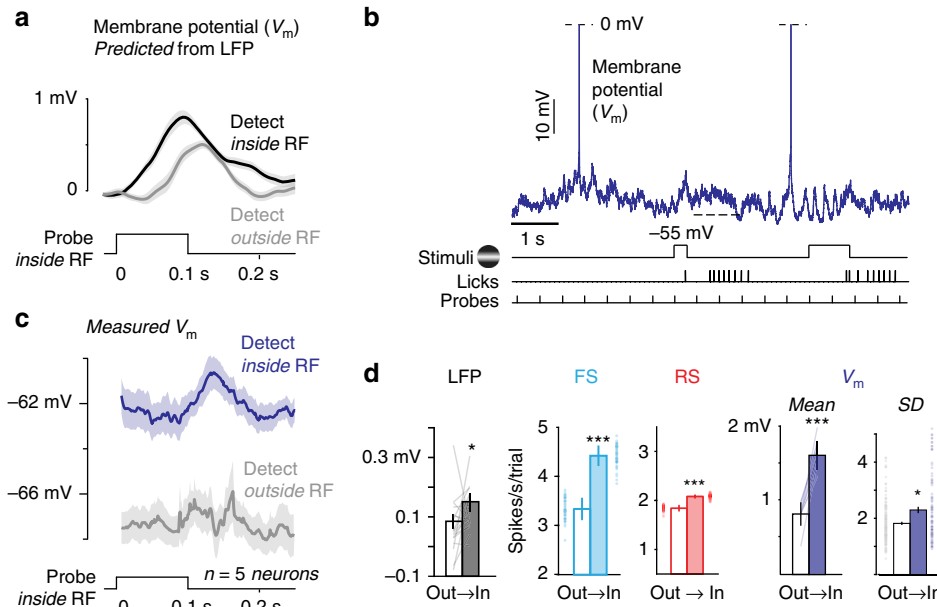

**Fig. 7 Selective enhancement of network, cellular, and subthreshold activity. a** Membrane potential ($V_m$) response predicted from LFP probe responses on detect inside (black) versus detect outside RF trials (gray). Mean ± s.e.m. Predictions generated from simultaneous LFP-Vm recordings outside of task (Supplementary Fig. 11). **b** Example whole-cell patch–clamp recording of $V_m$ during visual spatial detection. Spikes truncated at 0 mV. Time course of grating stimuli, correct detection (licks), and probe stimuli at bottom. **c** Membrane potential response is significantly more depolarized on trials with detection inside the RF (black, Δ2.2 ± 0.3 mV from 0.1 to 0.2 s, detect inside minus outside after DC subtraction, $p < 0.001$, signed-rank test, $n = 5$ neurons). **d** Spatial contextual modulation of probe-evoked LFP responses (79 ± 0.3%) significantly larger than for FS (32 ± 0.01%) or RS responses (13 ± 0.0004%; $p < 0.05$; Kruskal–Wallis nonparametric ANOVA; mean ± s.e.m.; all data recorded simultaneously from Trials$_{(4-6)}$, see Fig. 3; a single individual outlier was not plotted for LFP). LFP bar plots (Out vs In) values are 0.08 ± 0.02 mv (Out) and 0.15 ± 0.03 mV (In), $p < 0.05$ (Wilcoxon signed-rank test). Mean probe-evoked transient depolarization increased by 108 ± 20% ($p < 0.001$) and spontaneous SD $V_m$ increased by 23 ± 1.1% ($p = 0.01$). *$p < 0.05$, ***$p < 0.001$.

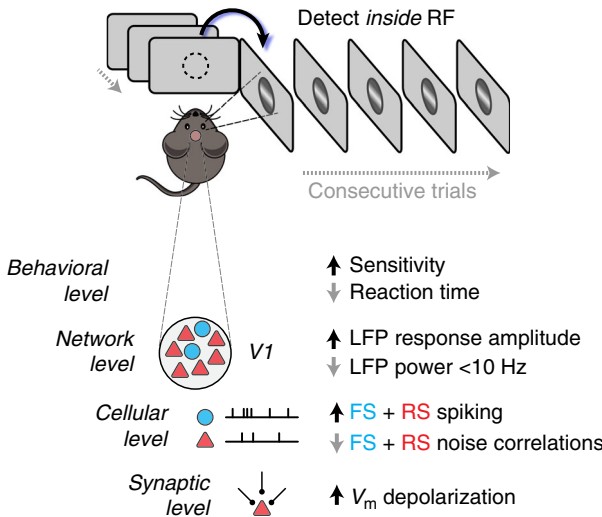

**Fig. 8 Summary of observed effects consistent with spatial attention.** At the behavioral level, detection speed and sensitivity improved. At the network level, LFP responses increased and <10 Hz power decreased. At the cellular level, RS and FS responses increased while noise correlations decreased. At the synaptic level, $V_m$ depolarized and $V_m$ fluctuations increased.

the RF. However, we could only assess spatial contextual modulation of neural responses inside versus outside the RF using probes, and these were brief and low contrast, sub-optimal conditions to evoke robust gamma activity[57]. Modifications to the basic task design and stimuli used here could drive gamma

activity more strongly and allow deeper examination of spatial contextual effects on spectral coherence, and between spikes and the LFP; indeed, we previously showed that the narrowband gamma oscillation in V1 driven by LGN[58] provided substantial predictive power for correct versus incorrect trials[12], deserving further investigation.

At the cellular level, detection inside the RF enhanced probe-evoked responses of putative excitatory and inhibitory neurons, while simultaneously decreasing their shared correlated variability. These findings are consistent with several studies of spatial attentional effects in both V1 and higher visual cortex of primates[20,21,25], and were not explained by several nonspatial or non-sensory factors. We found that spatial context most critically reduced correlated noise between RS and FS neurons. In mice, >90% of RS units are excitatory pyramidal cells, and nearly all FS units correspond to PV-positive inhibitory cells[59], providing greater confidence in identifying a cell-type-specific interaction. Decorrelation in FS and RS units is likely related to the reduction in low-frequency LFP power, since our previous study showed that low-frequency LFP activity directly relates to the amplitude of excitatory and inhibitory synaptic currents[44]. Unlike primate V4, we did not observe decreased variability of single-neuron spike statistics (e.g., Fano factor), but our results are consistent with a recent study of primate V1 that also did not observe Fano factor reduction with spatial attention[32]. In addition, we observed significant enhancement of both early and late components of probe-evoked spiking, whereas primate recordings typically observe attentional modulation of spiking and its variability at long latencies during stimulus presentation. In our study, spatial attentional modulation of probe responses inside the RF was most prominent in FS neurons, at long latencies (nearly 0.5 spikes per second greater late vs early). This

long-latency enhancement, although triggered by a task-irrelevant stimulus, reveals that a component of attentional modulation in mouse V1 is a temporally long-lasting increase in excitability of FS interneurons; this may be driven directly by neuromodulation of inhibitory neurons specifically with RFs at the spatial location where stimulus detectability improves across trials. Long-latency increases in excitability of both inhibitory and excitatory neurons following brief sensory responses have been observed in mice performing both visual and somatosensory tasks[60,61]. In our study, stimulation inside the RF was necessary to reveal these effects, since there was no long-latency increase in spiking following onset of blank stimuli; likewise, pupil measurements indicate that brain-wide arousal was not a major source for these effects, but it remains plausible that neuromodulation increases excitability specifically of V1 neurons only with RFs at the behaviorally relevant location, likely an important component of behavioral improvements with attention[62]. Our findings in mice enable techniques such as imaging of neuromodulatory axon terminals in the cortex to resolve this possibility[63]. These long-latency attentional effects appear different from observations in primates, which are primarily stimulus-driven (although there is evidence for attentional increases in spontaneous activity[40]). Moreover, attentional modulation of neural responses in primates occurs mainly during the sustained portion of the sensory response, not the initial sensory transient, as we have observed here across all levels of recording. The early component of feedforward sensory processing in primate sensory cortex may be relatively resistant to contextual modulation, whereas many studies in mice show that the earliest sensory signals in primary sensory cortex show clear modulation by motor, behavioral, and cognitive factors[3,4,6,8,10,14,60,61].

At the subthreshold level, we revealed that spatial context directly depolarizes cortical neurons with an accompanying increase in membrane potential variance. These first findings at subthreshold level, coupled with our observations at the cellular level, bolster long-standing theoretical predictions[18,27,28] that spatial attention operates through concerted increases of cortical excitation and inhibition that depolarize neurons via synaptic bombardment inside the receptive field. Rapid and transient depolarization to probes coupled with a small increase in spontaneous membrane potential variance during detection inside the RF supports models of gain modulation via increased synaptic activity[47,64], rather than models requiring decreased conductance and decreased membrane potential variance with no net depolarization[26]. Although measured in a small number of neurons, we observed both tonic and transient stimulus-evoked depolarization that was consistent with predictions calculated from the LFP responses, and also consistent with the level of probe-evoked spiking in RS neurons. Furthermore, our observation that subthreshold modulation is not solely restricted to the stimulus-evoked transient appears consistent with our observations of long-latency enhancement of RS and FS firing, the latter of which may be preferentially recruited to maintain stimulus selectivity[45] in the face of gain modulation via depolarization[47]; potential sources of sustained subthreshold depolarization may again arise from neuromodulatory inputs that increase neural excitability across trials as detection performance accelerates and improves. Identifying these and other subthreshold excitatory and inhibitory components of gain modulation[65] underlying spatial attentional effects now appears to be an approachable topic for future studies.

Our task used stimulus blocks and task-irrelevant probes to enable several key insights, but this also carries several inherent limitations. First, neither task-relevant gratings nor task-irrelevant probe stimuli were specifically tailored to drive optimal neural responses (e.g., best orientation and size), nor were probes

presented at a full range of test contrasts. Thus, the neuronal effects we measured are likely underestimated by combining across optimal and nonoptimal stimulus configurations; future studies could address this by embedding multiple-stimulus dimensions in the detected gratings and presenting probes at multiple contrasts. Second, several key predictions of theories of attention require cueing the locus of attention, and examining both detectability and feature selectivity, often with multiple competing stimuli[31]; based on our results, tasks with richer stimulus sets may now be feasible in mice, allowing testing of spatial attentional effects across large regions of the visual field, including those that exhibit unique spectral sensitivity[66]. Third, our current task design cannot systematically address the role of reward expectation and motivation across blocks, even if concurrent measures of pupil and licking suggest that arousal and motivation were not major factors that co-varied with behavioral and neural improvements. Future studies could parameterize reward probabilities and dissociate them from spatial context, then measure the specific contributions of each on behavior and on neural responses. Finally, our study established that several neural correlates of spatial contextual modulation appear in V1, but the sources require further investigation. Although these effects were most prominent in L4, they appeared across layers, suggesting sources could involve both subcortical[67,68] and higher cortical areas[22,69], where attentional effects may be even more pronounced[70–72]. Nonetheless, our results establish that mechanisms underlying improvements of behavior consistent with spatial attentional modulation may launch their effects from cells and synapses located at the very first stage of cortical sensory processing.

## Methods

**Mice, behavior, recordings.** Male C57BL6J (4–6 weeks old; reverse light cycle; all main Results) or PV-cre x Ai-32-ChR mice (Fig. 1c; Supplementary Fig. 1) learned to perform active visual detection of Gabor patches (horizontal orientation, 0.05–0.1 cycles per degree, phase randomized per trial) by licking for water rewards. Stimuli appeared only after mice withheld licking during a randomized interval on every trial (0.5 to 6 s, exponential distribution). Rewards were delivered upon first lick only during the stimulus window (typically 1–1.5 s). After 2–3 weeks, mice exhibited high-performance detection (d' >0.5) of small ($\sigma = 10°$) stimuli at multiple contrasts in both binocular and monocular visual space (10–25 consecutive trials per block per location), whereupon task-irrelevant probe stimuli were introduced during the intertrial interval (vertical bar 10° wide, ± 5% contrast, 0.1 s duration, 0.3 s interval, randomized location). We performed acute neural recordings with 32 site silicon probes or whole-cell patch–clamp recordings[45] from monocular V1 along with eye tracking (30 Hz) during behavior. Mice typically performed 4–8 consecutive blocks of interleaved monocular and binocular detection during recordings. All trained mice learned the task (no dropouts), and mice required V1 activity for behavioral performance (Supplementary Fig. 1). All procedures were approved by the Institutional Animal Care and Use Committee at the Georgia Institute of Technology and were in agreement with guidelines established by the National Institutes of Health and the Animals (Scientific Procedures) Act 1986 (UK).

**Surgery.** Male C57BL6J mice (4–6 weeks old; reverse light cycle housing; bred in house; IMSR_JAX:000664) or offspring of PV-cre (IMSR_JAX:017320) crossed with Ai-32-ChR (IMSR_JAX:024109) mice were surgically implanted with a stainless steel headplate (with 3–4 mm diameter recording chamber) affixed to the skull over primary visual cortex (V1) using a thin layer of veterinary adhesive (VetBond) and biocompatible polymer (Metabond). Following implantation and recovery (3 days), animals were handled and acclimatized for 3–4 days to the head fixation apparatus and placed under a restricted water schedule. Mice received a fixed daily amount of water scaled for body weight (40 ml per kg per day; 1 ml per day for a typical 25 g mouse), according to established methods[12].

**Behavioral training.** Mice first learned the contingency between stimulus and water reward through instrumental conditioning. In these sessions, reward was automatically delivered after a fixed delay (0.6–0.7 s) relative to stimulus onset. We monitored the first lick latency for precession from reward delivery to visual stimulus onset (anticipatory licking). We then transitioned to active visual detection, where reward delivery occurred only if the mouse licked during the stimulus window (typically 1.5–2 s early in training). The stimulus disappeared and reward was delivered upon first lick during the response window. Detection performance was quantified with signal-detection theory metrics (d-prime, d'). Hit rates were

calculated from correct detection trials (licks during stimulus window), while false alarm rates were calculated from trials with blank stimuli (0% contrast; 20% of trials). Trials with stimulus presentation but no licks or late licks were classified as Miss trials. When d' was above chance levels for 2 consecutive days, stimulus contrast range and/or size was decreased to maintain difficulty. Once animals exhibited performance above chance for both binocular and monocular stimuli of high and low contrast, we introduced task-irrelevant probe stimuli, described below. In well-trained mice, these faint and brief bars did not affect behavioral performance and caused no change in licking frequency above baseline (Supplementary Fig. 5).

**Stimulus properties**. Mice detected static horizontal Gabor gratings, presented in either the binocular (central ±25° from vertical meridian) or monocular (60°–90° from vertical meridian) visual field on linearized LCD monitors (60 or 80 Hz refresh rate) with isoluminant 50% gray background. Grating phase was randomized per trial, while spatial frequency (range 0.05–0.1 cycles per deg) and stimulus size (σ range 10°–20°) remained fixed across blocks of trials. Binocular contrasts ranged from 2% to 75%, monocular contrasts ranged from 7% to 90%. Responses at all contrasts within spatial location were combined for main figures. Probe stimuli (10°-wide vertical bars, black or white) were briefly presented (0.1 s duration, 0.3 s interval) at low contrast (±5% from grey background) one at a time in unique and randomly selected spatial locations (1-D white noise). These locations tiled the entire visual field (16 discrete locations, spanning 145° of the visual hemifield, Supplementary Fig. 4). Thus, only a small fraction of probes were presented inside the receptive field (RF), and these were much lower contrast than the grating stimuli. We aggregated responses of the three central most probe positions inside the RF for analysis (central ± 1; Supplementary Fig. 4). Note that RFs in mouse V1 are ~30° in diameter[1,45] (Supplementary Fig. 4; Supplementary Fig. 11), so these three adjacent probes are within the centre of the RF.

**Cortical inactivation**. Optogenetic inactivation experiments with PV-cre x Ai-32 (ChR2) mice (Fig.1; Supplementary Fig. 1) were carried out as in our prior study[12]. The skull was thinned over monocular V1 and a fiber-coupled LED delivered pulses of blue light (473 nm; 4.1–6.5 mW at the cranium) over monocular V1 (subsequently confirmed with neural recordings) on 25% of detection trials (1 s during visual stimulus, ramping from 0.1 s before stimulus). Similar deficits of task performance were also observed with pharmacological inactivation of V1[12].

**Recordings**. On the day of recording, a small (~100–500 μm) craniotomy was made over monocular V1 (0.5 mm anterior to lambda, 2–2.5 mm lateral to central suture) under isoflurane anesthesia. Mice recovered >3 h before recording sessions. We used multisite silicon probes (NeuroNexus) consisting of either a single 32-channel shank, or two 16-channel shanks. Electrodes were advanced ~ 1000 μm below the cortical surface. Raw electrical signals were amplified and digitized (Blackrock Microsystems) then exported for post processing. At the end of behavior, we performed receptive field mapping of the recording site (100% contrast vertical flashed bars, 10° width, duration 0.1 s, interstimulus interval 0.3 s, placed in random locations tiling 144° of the visual field). The craniotomy was covered with elastomer in between consecutive recording days (typically 2–4 from the same site). Whole-cell patch–clamp recordings were performed in L2/3 as in our prior studies[44,45]. In a subset of mice, we simultaneously recorded pupil with a high speed camera and infrared illumination[12].

**Spike sorting and LFP analysis**. Extracellular spikes were isolated using the KlustaViewa Suite[73], as detailed in our prior study[12]. Briefly, automated clustering was followed by manual curation involving (1) removal of obvious noise clusters, (2) classification of poorly isolated waveforms as multiunit activity, and (3) PCA space curation in parallel with unit auto- and cross-correlation histogram analysis for single unit identification. Histograms of peak-to-trough spike widths were bimodal (Supplementary Fig. 6); FS neurons had width <0.57 ms, with broader units classified as RS. This classification agrees with several studies of mouse V1[1,38], where FS neurons consist nearly exclusively of parvalbumin (PV)-positive inhibitory neurons[59]. In our experiments, narrow spike widths and high firing rates of FS neurons correspond closely with spikes optogenetically activated in PV interneurons[12]. Peri-stimulus histograms (PSTHs) for each cell type were constructed by binning the population responses from each recording in 25-ms bins. Population PSTHs were smoothed ± three bins for visualization.

LFP was isolated by bandpass filtering raw neural signals from 0.3 to 200 Hz. Laminar profiles were calculated by using current source density analysis[1,12,38], with laminar boundaries consistent with prior studies[74]. For all stimulus-evoked analyses, the mean LFP preceding stimulus presentation was subtracted from each individual LFP response. These were then averaged across all stimulus presentations per condition (e.g., probe responses inside the RF on detect inside RF trials), and then averaged across recording sessions and mice.

LFP power spectra were computed using Welch's power spectral density estimate (MATLAB "pwelch" function). Modulation indices (MI) were computed by comparing the integrated power from 0 to 10 Hz on trial 1 vs trial 5:

$$\text{MI} = \frac{\text{Trial } 5 - \text{Trial } 1}{\text{Trial } 5 + \text{Trial } 1} \qquad (1)$$

**Data analysis and statistics**. Neural data analysis was carried out only on correct detection trials (21 sessions, 15 mice), or in non-task conditions (7 sessions, 4 mice; Supplementary Fig. 9). Broadband signals were separated into LFP (0.3–200 Hz) and spikes[73] of FS ($n = 106$) putative inhibitory and regular spiking (RS, $n = 306$) putative excitatory units (Supplementary Fig. 6). Receptive fields were mapped for all recordings, responses to probes in the center ± 1 locations inside the RF were combined (Supplementary Fig. 4). Noise correlations were computed as in our prior study[12]. All data analysis utilized MATLAB, and statistical comparisons used nonparametric Wilcoxon rank-sum tests (unpaired data), or signed-rank tests (paired data) unless otherwise noted, with p-values indicated in figure legends or text where appropriate. The data structures and code that support the findings of this study are available from the corresponding author upon reasonable request.

**Correlations**. Noise correlations were calculated between smoothed spike trains (20-ms Gaussian filter) by subtracting the mean probe response for each neuron and then computing cross correlations at all lags (MATLAB "xcorr" function with "coeff" normalization). Probe response spike trains consisted of the 0.1 s preceding the probe onset until the onset of the next probe stimulus. Only probes within 10° of the receptive field center were considered. Statistical comparisons examined peak correlation values.

**Pupil analysis**. Pupil analysis was performed as in prior studies[12]. Briefly, area changes were calculated per trial as the percent deviation from the mean area across the entire session, and position was analyzed for changes in the azimuth (since stimulus position across binocular and monocular blocks varied in azimuth).

**Gain modulation analysis**. Probes of multiple contrast were presented in awake mice outside of the behavioral task. Responses were used to construct contrast response functions (CRFs, Supplementary Fig. 10). Colored circles show two response types measured during the task (1) the circle at 5% contrast shows the mean population response to probes during detection trials inside the RF, (2) two circles at high contrasts show the mean population response to gratings at these contrasts. These in task responses (colored circles) were not used for contrast response curve fits for either model. Rather, the control CRF constructed from probes outside of the task was either (a) shifted left (contrast gain) to capture enhancements of the 5% contrast probes during detection trials inside the RF or (b) multiplied (response gain) to capture the same enhancements to same 5% probes during detection trials inside the RF. The shifted or multiplied curves were then used to predict the amplitude of the high contrast grating responses. Errors were calculated by measuring the % difference as defined below at each trial for each contrast:

$$\%\text{Difference} = \frac{\text{predicted} - \text{measured}}{\text{predicted}} \times 100 \qquad (2)$$

Errors were then averaged across trials and across recordings.

**Reporting summary**. Further information on research design is available in the Nature Research Reporting Summary linked to this article.

## Data availability

The data structures and code that support the main findings of this study are available from the corresponding author upon reasonable request. Data will also be made available from the corresponding author's institutional website (https://haider.gatech.edu/).

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

## Acknowledgements

We thank Alexander Zorn and Hayley Arrowood for technical assistance, Aman Saleem and Chris Burgess for advice on behavior, James Mazer and Dobromir Rahnev for comments on the paper, and Matteo Carandini for support in the initial phase of the study. J.D.R. was funded by Goizueta Foundation and Sloan Foundation, B.H. was funded by the Whitehall Foundation, Sloan Foundation, GT Neural Engineering Center, NIH NINDS (1R01NS107968), and NIH BRAIN Initiative (1R01NS109978).

## Author contributions

A.S., J.D.R. and B.H. performed experiments; A.S., B.H., J.D.R. and N.M. performed data analysis; B.H. designed the study; B.H., A.S. and J.D.R. wrote the paper.

## Competing interests

The authors declare no competing interests.
