## [Peer Review File · Nature Communications]

Reviewers' Comments:

Reviewer #1:

Remarks to the Author:

The authors have addressed many of the issues I previously raised. The new plots of false alarm rates and sensitivity improve the presentation substantially. I strongly recommend adding the first R1 figure (R1-5) to the manuscript, ideally in the main text. These data on behavior across contrasts greatly improves the characterization of what the animals are doing, and will be valuable to readers.

One point deserves further discussion. In R1-8, the authors say that stimulus-evoked activity continues after the stimulus is extinguished. The more critical point is that the spatially selective activity `_enhancement_` appears to continue after the stimulus is extinguished. This is a difference from primate studies of spatial attention which often report a change in gain, that is, a change in the stimulus-evoked spike rate response. And the issue is important given how the authors compare mouse and primate models of attention. While the data shown makes it a little hard for me to judge how strong the effect is, this topic should be addressed in the text, and presenting a panel of data in the text/supp info might also be useful.

Reviewer #2:

Remarks to the Author:

The authors improved some aspects of this manuscript. Specifically, careful rewording of the task is an improvement as is inclusion of more detailed information about the task and probe stimuli. Unfortunately, the major concerns raised by multiple reviewers were not adequately addressed in this revision.

The first major concern, raised by all three reviewers, was that this work is largely replication of prior studies with minimal new insight. The only findings that have not been reported previously in a similar task context are the membrane potential results, but these are from such a small sample that interpretation of results is challenging (furthermore, a simple bar graph is not appropriate for display of results from such a small sample). Additionally, multiple reviewers commented on the superficiality of the findings, suggesting that more data be obtained in order to make stronger and deeper claims about the system. As additional data were not included to support the main claims of the paper, this work does not seem ready for publication.

The second major concern, also raised by multiple reviewers, was that the observed neurophysiological changes could be better explained by changes in expectation or motivation rather than visually evoked responses. All reviewers commented on the fact that neuronal responses do not appear to be entirely visually-driven, i.e. responses continue to ramp up for 100's of msec after visual stimuli are off. The argument provided by the authors that visual latencies are different in mice does not account for continued increases in firing rate 100's of msec after stimulus offset. This non-visual response seems more consistent with an expectation signal than something related to visual spatial position. Finally, the new behavioral data do not satisfy the concern about the source of the neurophysiological "ramping" effects. The authors noted that mice respond more across trials within a block – does this not suggest an increase in expectation or motivation?

In addition to not addressing the major concerns laid out in the first round of review, there are new concerns raised by some of the authors' responses.

1. In the R1-5 figure, behavior across stimuli varying in contrast are illustrated for 3 mice – why were data only shown for these 3 mice? What about the other 12 mice? This also raises a concern about which mice were used for which experiments. Were the same 15 mice used for all behavioral and neurophysiological assessments? This should be very clearly spelled out in the main text.
2. The probe stimuli were very large (9 degree bars). It is still not clear whether the presence of the probe stimuli altered performance. Specifically, what happened on trials when the probe and the stimulus to detect were both displayed? According to Figure 2a and the text, this happened frequently and yet it is not clear how the R2-19 figure addresses this condition. Furthermore, LFP responses to probes plus gratings appear to be different from LFP responses to gratings alone – these differences are larger than those illustrated in Figure 1. It is still not clear how the probe stimuli impact behavior and neurophysiological responses, especially when combined with visual stimuli.

We thank the Reviewers for their additional comments that have provided constructive guidance for expanding the scope and rigor of our revised manuscript.

We have now integrated all of the Reviewer figures as additions to the paper, and in several instances have elevated supplemental figures to main text figures.

By including these previous analyses, and providing new analyses in response to Reviewers, we have greatly expanded the scope of the original work (8 Main figures) and present a more comprehensive, careful, and considered analysis of neural modulations arising from spatial context in mouse visual cortex.

Specific additions to revised manuscript. The revised manuscript now comprehensively describes the effects of spatial context across behavioral (Fig. 1), network (Figs. 1 & 3), cellular (Fig. 3 & 5), and subthreshold (Fig. 7) levels. These results now live side-by-side with much of the analysis suggested by Reviewers to assess neural effects of arousal (Fig. 2), motivation (Fig. S3), and stimulus expectation (Figs. S7-8). We have accordingly expanded the Results and Discussion sections, providing further depth for our study.

Substantially new text in the manuscript is show in blue.

Specific responses to reviewer concerns follow below.

Reviewer #1 (Remarks to the Author):

(R1)-1: The authors have addressed many of the issues I previously raised. The new plots of false alarm rates and sensitivity improve the presentation substantially. I strongly recommend adding the first R1 figure (R1-5) to the manuscript, ideally in the main text. These data on behavior across contrasts greatly improves the characterization of what the animals are doing, and will be valuable to readers.

We thank the Reviewer for this suggestion. We have now added an improved version of this figure (including hit rate, d' , and psychometric curves) to Suppl. Fig. 1. The new figure is shown below.

(R1)- 2. One point deserves further discussion. In R1-8, the authors say that stimulus-evoked activity continues after the stimulus is extinguished. The more critical point is that the spatially selective activity_enhancement_ appears to continue after the stimulus is extinguished. This is a difference from primate studies of spatial attention which often report a change in gain, that is, a change in the stimulus-evoked spike rate response. And the issue is important given how the authors compare mouse and primate models of attention. While the data shown makes it a little

hard for me to judge how strong the effect is, this topic should be addressed in the text, and presenting a panel of data in the text/supp info might also be useful.

We have now compared the effects of the Early and Late components modulation to the probe stimulus, and find that there is a slight but statistically significant increase in RS and FS responses (FS: Early, 4.43 ± 0.21 spikes/s vs Late, 4.75 ± 0.18 , $p < 0.05$; RS: Early, 2.08 ± 0.05 vs Late, 2.13 ± 0.07 , $p < 0.05$).

We have now integrated this result into Fig. 3 depicting the results of Early vs Late probe stimulus response modulation, and address this in the text (p. 6, l. 156).

Reviewer #2 (Remarks to the Author):

The authors improved some aspects of this manuscript. Specifically, careful rewording of the task is an improvement as is inclusion of more detailed information about the task and probe stimuli. Unfortunately, the major concerns raised by multiple reviewers were not adequately addressed in this revision.

(R2-1): The first major concern, raised by all three reviewers, was that this work is largely replication of prior studies with minimal new insight. The only findings that have not been reported previously in a similar task context are the membrane potential results, but these are from such a small sample that interpretation of results is challenging (furthermore, a simple bar graph is not appropriate for display of results from such a small sample).

We appreciate the Reviewer's point questioning the level of insight provided by replication. In fact, the Reviewer's criticism rests on accepting that our pioneering observations in mice are consistent with results from multiple visual brain areas of primates. For this very reason, we feel that a manuscript that convincingly establishes cross-species consistency and replication of spatial contextual modulations in mouse V1 provides both novelty and a major step forward for studies in mice.

We reiterate: there are no prior reports describing any one of our multiple observations of neural modulation by spatial context in the mouse visual system. Our report provides bedrock for future studies that will probe mechanisms and circuits underlying spatial contextual modulation, using tools currently only available in mice.

Moreover, our novel findings are not merely limited to the membrane potential results, which represent a minor component of our assembled findings in the revised manuscript. The integrity and originality of our study does not depend solely upon the membrane potential findings.

Our report establishes at least 10 novel results in mice, spanning multiple scales:

Behavioral level:

1. Improvements in both sensitivity and reaction time (Fig. 1) on trials with simultaneous neural modulation in V1 (Figs. 1, 3, 5 - 7)
2. Behavioral improvements unexplained by eye movements (Fig. 2), increased arousal (Fig. 2), or by changes in motivation for reward (Fig. S3)

Network level

3. Larger LFP responses in L4 of V1 to stimuli detected across successive trials (Fig. 1)

4. Larger LFP responses to probe stimuli inside the RF, specifically on detection trials inside the RF (Fig. 3)
5. Suppression of low frequency LFP fluctuations in L4 and L5/6, specifically on detection trials inside the RF (Fig. 6)
6. LFP response enhancement unexplained by arousal (Fig. 2) or stimulus expectation (Figs. S7-8).

Cellular level:

7. RS and FS response enhancement to probe stimuli inside the RF, specifically on detection trials inside the RF (Fig. 3)
8. Reduction in noise correlations among RS-FS pairs, specifically on detection trials inside the RF (Fig. 5)
9. No evidence for linearly increasing firing rates due to stimulus expectation (Figs. S7-8).

Subthreshold level:

10. Spatially specific depolarization and increased membrane potential fluctuations during detection inside the RF (Fig. 7).

To provide a clear visual depiction of these novel findings to readers, we now provide a summary figure (Fig. 8) that illustrates our major results across these multiple scales.

(R2-2): Additionally, multiple reviewers commented on the superficiality of the findings, suggesting that more data be obtained in order to make stronger and deeper claims about the system. As additional data were not included to support the main claims of the paper, this work does not seem ready for publication.

In the comments to authors, no Reviewer explicitly requested the addition of experimental data to address or resolve specific interpretational concerns. Reviewer 1 specifically requested behavioral analysis of sensitivity and false alarms, which we added in the previous revision (and Fig 1 here), and which the Reviewer stated “*improves the presentation substantially.*”

In response to additional specific critiques, we have provided substantial new analysis of data to provide an even more comprehensive picture of the neural modulations induced by spatial context in the mouse visual cortex. (5 new Main Figures; Results Listed in R2-1).

Furthermore, inspired by the Reviewer’s prior critiques regarding spectral modulation of the LFP, we have now performed analysis of LFP spectral power on early versus late trials of detection inside the RF (Trials 1 versus 5, identical to all other analyses in the manuscript).

We find that detection inside the RF significantly reduce LFP power in frequencies <10 Hz on Trial 5 versus Trial 1. These reductions were not only present in L4 (the major input layer of V1, and layer of interest in our report), but also in output Layers 5/6.

These results establish that in mice, salient spatial context reduces low frequency fluctuations, establishing further consistency with effects of spatial attention in primate visual cortex (Mitchell et al., 2009; Chalk et al., 2010; Khayat et al., 2010)

We have added this Result as Fig. 6.

In addition, inspired by the Reviewer's critiques, we have included substantial new analysis of zero contrast trials to address concerns about non-sensory "ramping" modulation (See R2-3, below).

Lastly, as the Reviewer requested, we have added the individual data points to all histograms in Fig. 3 and Fig. 7 that summarize neural response modulations across LFP, RS, FS, and V_m . These data show statistically significant neural modulations across several levels of neural recording (just as the prior version did).

(R2-3): The second major concern, also raised by multiple reviewers, was that the observed neurophysiological changes could be better explained by changes in expectation or motivation rather than visually evoked responses. All reviewers commented on the fact that neuronal responses do not appear to be entirely visually-driven, i.e. responses continue to ramp up for 100's of msec after visual stimuli are off. The argument provided by the authors that visual latencies are different in mice does not account for continued increases in firing rate 100's of msec after stimulus offset. This non-visual response seems more consistent with an expectation signal than something related to visual spatial position. Finally, the new behavioral data do not satisfy the concern about the source of the neurophysiological "ramping" effects.

We appreciate the Reviewer's insistence for further analysis identifying if expectation and motivation underlie non-sensory ramping of neural responses.

We have now performed extensive analysis of 0% contrast (blank) stimulus trials. Blank trials comprised 20% of the total, were randomly interleaved throughout the behavioral sessions, and occurred with the same randomized inter-trial intervals as the grating stimuli detected for rewards (see Methods). This allowed us to isolate firing rates both preceding and immediately following the "expected" appearance of a visual stimulus across trial blocks, but in the absence of any sensory-evoked activity. *Indeed, there was no increase in firing rates across the population of RS and FS neurons preceding or immediately following the onset of a blank stimulus on first or fifth trials within block.*

Analyses of blank stimuli shows that **(1)** stimulus expectation does not produce non-sensory "ramping" of neural activity across trials, and **(2)** does not produce "ramping" in a spatially-specific manner.

1. To address concerns about modulation due to stimulus expectation across trials, we calculated average neural activity in RS neurons (left columns, red) and FS neurons (right columns, cyan), segregated by trials early in the block (grey) versus late in the block (colored), exactly as analyzed in the main Results. There was no evidence of significant positive increase of firing rate ("ramp") in anticipation of

“expected” stimulus onset, for any comparison (best fit lines and p values for linear correlation indicated in each subplot).

Blank trials also allowed us to separately analyze trials where a lick was produced in the absence of the sensory stimulus (False alarms, A and C), versus trials where no lick was produced (“Correct reject”, B and D). Again, there was no indication that an impending lick produced non-sensory “ramping” of firing rates early or late in the block.

2. To address concerns about modulation due to stimulus expectation in discrete spatial locations, we carried out a similar analysis of blank trials, but segregated them according to trials of detection inside versus outside the RF. As shown at right, there was no evidence of significant positive increase of firing rate (“ramp”) in anticipation of “expected” stimulus onset inside the RF, for any comparison (colored traces as above, p values indicated).

Only one comparison provided evidence for a positive increase of firing rates on blank trials, but this was for FS cells on detection trials outside the RF (not inside the RF, the major source of modulation in our manuscript).

This single instance of modulation was only apparent on false alarm trials when a lick was eventually produced (Figure, C); trials where no lick was produced showed no increase in firing before stimulus onset (Figure, D). Nonetheless, these results show that in no case did detection inside the RF produce evidence for non-sensory “ramping” of firing rates.

These important analyses inspired by the Reviewer’s critiques are now included as Suppl. Fig 7 and 8.

(R2-4) The authors noted that mice respond more across trials within a block – does this not suggest an increase in expectation or motivation?

The Reviewer correctly notes that mice responded more across trials within a block; importantly, more frequent responses were driven by more Hit trials, and fewer false alarms.

We reasoned that changes in motivation across trials would manifest as measurable changes in licking. If motivation for reward increased from first to fifth trials, this should drive impulsivity and increase premature licks preceding stimulus onset; likewise, consummatory licking should speed up across sequential rewards if motivation grew. *In fact, there was no change in the fraction of trials with anticipatory stray licks across the block of trials, suggesting that mice are not licking more frequently in anticipation of the stimulus across trials* (Figure, A).

To assess if motivation for reward was increasing across trials, we measured the number of licks produced during Hit trial responses. These were essentially flat across trials, suggesting a constant level of motivation for obtaining and consuming reward across trials. (Figure, B)

These two figures have been included as Suppl Fig. 3.

(R2-4): In addition to not addressing the major concerns laid out in the first round of review, there are new concerns raised by some of the authors' responses.

In the R1-5 figure, behavior across stimuli varying in contrast are illustrated for 3 mice – why were data only shown for these 3 mice? What about the other 12 mice? This also raises a concern about which mice were used for which experiments. Were the same 15 mice used for all behavioral and neurophysiological assessments? This should be very clearly spelled out in the main text.

The additional behavioral data were shown specifically for these 3 mice because they were trained with the same contrast range for several weeks. These 3 mice were included for both behavioral and neural data analysis, and are now shown in Fig. S1.

Furthermore, these 3 mice plus the other 12 mice used for behavioral analysis were exactly the same population analyzed during neural recordings, strengthening the concordance of our results. We have added a sentence clarifying this at the outset of the results (p. 4, l. 95).

(R3-4): The probe stimuli were very large (9 degree bars). It is still not clear whether the presence of the probe stimuli altered performance. Specifically, what happened on trials when the probe and the stimulus to detect were both displayed? According to Figure 2a and the text, this happened frequently and yet it is not clear how the R2-19 figure addresses this condition.

The previous figure R2-19 addressed the increase in false alarms due to probes inside the RF. The Reviewer is correct that this does not explicitly address the effect of probes during grating presentation. We now show that Hit rates were indistinguishable on trials where probes co-occurred with the grating, versus trials where they did not appear with the grating. The figure below is now included as Suppl. Fig. 5, and establishes that neither false alarm licks nor correct detection licks were impacted by the probe stimuli.

(R3-5): Furthermore, LFP responses to probes plus gratings appear to be different from LFP responses to gratings alone – these differences are larger than those illustrated in Figure 1.

It is still not clear how the probe stimuli impact behavior and neurophysiological responses, especially when combined with visual stimuli.

Please see above figure for analysis of Hit rates with probes plus gratings.

The Reviewer states that neural responses to probes plus gratings “are larger than those illustrated in Figure 1.” This was not the case. The early-stimulus response (depth-negative LFP) to probes plus gratings was not significantly different from grating alone responses, while the modulation of grating responses across trials was 75% larger than this. We re-plot all conditions below for side-by-side comparison.

These panels are now included in Suppl. Fig. 5.

Reviewers' Comments:

Reviewer #1:

Remarks to the Author:

I am positive about this work. I do not share the first major concern of R2. I agree with the authors that reporting this result in mice is novel and an important contribution.

However, R2's second concern in this round of review is the same as my point in this round ("One point deserves further discussion..."). The present revision does not fully address the issue, but I think that merely text changes are enough to address it. The authors should add substantial text, 5 sentences at least, in Results and Discussion, to elaborate on Fig. 3cd. Specifically, the authors should note that the Vm attentional modulation is not restricted to the time the stimulus is on, and occurs before and substantially after the stimulus. This is almost certainly a signature of some kind of temporally long-lasting attentional or arousal effect, and it is different than what attention does in primates. The authors should be unafraid to admit this may be an arousal effect, and marshal the arguments for and against such an interpretation in their text. The issue is important enough to require some expansion of Results and Discussion, and I might even mention it in the abstract.

The manuscript is strengthened by the various additional analyses added in this revision.

Finally, please put the reviewer figure (now R1-1, was R1-5) into a main figure, not just add it as a supplementary panel.

Response to Reviewers

Reviewer #1 (Remarks to the Author):

I am positive about this work. I do not share the first major concern of R2. I agree with the authors that reporting this result in mice is novel and an important contribution.

We thank the Reviewer for his positive and supportive view of the revised manuscript.

1. *However, R2's second concern in this round of review is the same as my point in this round ("One point deserves further discussion..."). The present revision does not fully address the issue, but I think that merely text changes are enough to address it. The authors should add substantial text, 5 sentences at least, in Results and Discussion, to elaborate on Fig. 3cd. Specifically, the authors should note that the Vm attentional modulation is not restricted to the time the stimulus is on, and occurs before and substantially after the stimulus. This is almost certainly a signature of some kind of temporally long-lasting attentional or arousal effect, and it is different than what attention does in primates. The authors should be unafraid to admit this may be an arousal effect, and marshal the arguments for and against such an interpretation in their text. The issue is important enough to require some expansion of Results and Discussion, and I might even mention it in the abstract.*

1. The Reviewer's suggestion to highlight that "*attentional modulation is not restricted to the time the stimulus is on*" is important. We have now included language very close to the Reviewer's suggestions, and have developed the interpretation of evidence for and against long-lasting effects. We reproduce this new text below:

RESULTS (1 sentence added, following description of Fig. 3c-d):

(p. 6, line 143): Like the LFP, sensory response enhancements were prominent at short latency (<0.15 s) and slightly but significantly larger at longer latencies (0.15 – 0.3s), perhaps owing to delayed and prolonged responses to low stimulus contrast³⁹, or to spatially specific but sustained increases in neural excitability resulting from improved behavioral performance across trials.

DISCUSSION (10 sentences added):

(p. 11, l. 290): In our study, spatial attentional modulation of probe responses inside the RF was most prominent in FS neurons, at long latencies (nearly 0.5 spikes per s greater late vs. early). This long-latency enhancement, although triggered by a task-irrelevant stimulus, reveals that a component of attentional modulation in mouse V1 is a temporally long-lasting increase in excitability of FS interneurons; this may be driven directly by neuromodulation of inhibitory neurons specifically with RFs at the spatial location where stimulus detectability improves across trials. Long-latency increases in excitability of both inhibitory and excitatory neurons following brief sensory responses have been observed in mice performing both visual and somatosensory tasks^{60,61}. In our study, stimulation inside the RF was necessary to reveal these effects, since there was no long-latency increase in spiking following onset of blank stimuli; likewise, pupil measurements indicate that brain-wide arousal was not a major source for these effects, but it remains plausible that neuromodulation increases excitability specifically of V1 neurons only with RFs at the behaviorally relevant location, likely an important component of behavioral improvements with attention⁶². Our findings in mice enable techniques such as imaging of neuromodulatory axon terminals in cortex to resolve this possibility⁶³. These long latency attentional effects appear different from observations in primates, which are primarily stimulus driven (although there is evidence for attentional increases in spontaneous activity⁴⁰). Moreover, attentional modulation of neural responses in primates occurs mainly during the sustained portion of the sensory response, not the initial sensory transient, as we have observed here across all levels of recording.

(p.12, line 327): Furthermore, our observation that subthreshold modulation is not solely restricted to the stimulus-evoked transient appears consistent with our observations of long-latency enhancement of FS firing, which may be preferentially recruited to maintain stimulus selectivity⁴⁵ in the face of gain modulation via depolarization⁴⁷.

2. The manuscript is strengthened by the various additional analyses added in this revision. Finally, please put the reviewer figure (now R1-1, was R1-5) into a main figure, not just add it as a supplementary panel.

2. The two main panels of R1-1 are now included in Fig. 1b-c, as shown below.

The original R1-1 (also showing hit rate curves and binocular performance with inactivation) is still shown in Supplementary Figure 1.